# Idiopathic Normal-Pressure Hydrocephalus Revealed by Systemic Infection: Clinical Observations of Two Cases

**DOI:** 10.3390/neurolint17060086

**Published:** 2025-05-30

**Authors:** Shinya Watanabe, Yasushi Shibata, Kosuke Baba, Yuhei Kuriyama, Eiichi Ishikawa

**Affiliations:** 1Department of Neurosurgery, Mito Kyodo General Hospital, Tsukuba University Hospital Mito Area Medical Education Center, Mito 310-0015, Japan; yshibata@md.tsukuba.ac.jp (Y.S.); kosuke.baba6211@gmail.com (K.B.); yuhei.19970221@gmail.com (Y.K.); 2Institute of Medicine, University of Tsukuba, Tsukuba 305-8576, Japan; e-ishikawa@md.tsukuba.ac.jp

**Keywords:** idiopathic normal pressure hydrocephalus, sepsis, cerebrospinal fluid, ventriculoperitoneal shunt, sepsis-associated encephalopathy, gait disturbance

## Abstract

**Background/Objectives:** Idiopathic normal-pressure hydrocephalus (iNPH) is a potentially reversible neurological disorder characterized by gait disturbance, cognitive impairment, and urinary incontinence. Its pathophysiology involves impaired cerebrospinal fluid (CSF) absorption, and recent research has highlighted the role of the glymphatic and meningeal lymphatic systems in this process. However, the factors that trigger the clinical manifestations of iNPH in subclinical cases remain poorly understood. **Case Presentation**: Herein, we report two rare cases of iNPH in which clinical symptoms only became apparent following systemic infections. An 82-year-old man presented with transient neurological deficits during a course of sepsis caused by *Klebsiella pneumoniae*. Neuroimaging revealed periventricular changes and mild ventricular enlargement. Shunting and a tap test led to significant improvements to both his gait and cognition. An 80-year-old man with a history of progressive gait disturbance and cognitive decline developed worsening urinary incontinence and acute cerebral infarction caused by *Staphylococcus haemolyticus* bacteremia. Magnetic resonance imaging revealed a ventriculomegaly with features of disproportionally enlarged subarachnoid space hydrocephalus and a corona radiata infarct. Clinical improvement was achieved after a ventriculoperitoneal shunt was placed. **Conclusions**: Our two present cases suggest that systemic inflammatory states may act as catalysts for the manifestation of iNPH in patients with predisposing cerebral ischemia or subclinical abnormalities in CSF flow, highlighting the need for higher clinical awareness of iNPH in older patients who present with neurological deterioration during systemic infections. Early diagnosis and timely shunting after appropriate infection control may facilitate significant functional recovery in such patients.

## 1. Introduction

Idiopathic normal-pressure hydrocephalus (iNPH) is a condition characterized by gait disturbances, cognitive impairment, and urinary incontinence in the absence of precipitating events such as subarachnoid hemorrhage or meningitis [1]. It is thought to result from impaired absorption of cerebrospinal fluid (CSF) in the brain. The current understanding of CSF absorption pathways has evolved significantly over recent years [2,3,4,5,6]. Recent studies have highlighted the role of the glymphatic system in CSF circulation and waste clearance. Impaired glymphatic function has been implicated in the pathogenesis of iNPH [7,8]. These findings further support our hypothesis that infection-related inflammation may interfere with CSF homeostasis, particularly in individuals with pre-existing subclinical dysfunction. The glymphatic system and intra-arterial intramedullary drainage pathways have thus emerged as the major contributors to CSF clearance [2]. These insights provide a more nuanced view of the pathophysiology of hydrocephalus. Some patients with untreated iNPH show a gradual reduction in ventricular forward flow that parallels the worsening of their clinical symptoms and can eventually lead to irreversible progressive ischemic damage in the brain [9].

Herein, we report two cases of iNPH in which the clinical symptoms were complicated by progressive cerebral ischemic damage and became apparent only following systemic infections. In one case, the patient presented with transient neurological symptoms that mimicked cerebrovascular disease during a course of sepsis, and the second appeared to resemble cerebral infarction. Both cases suggest that infection-related disturbances to CSF dynamics may have triggered the overt manifestations of underlying iNPH.

## 2. Case Presentation

### 2.1. Case 1

An 82-year-old man presented to our center with a one-week history of transient generalized weakness followed by fever beginning four days prior to admission. He was initially evaluated at another hospital, where he exhibited lower limb weakness. Head computed tomography (CT) revealed bilateral periventricular hypodensities. His symptoms subsequently improved, and he was transferred to our institution for further evaluation.

His medical history included hypertension, dyslipidemia, and diabetes mellitus. He was taking medications for these conditions and had no history of smoking or alcohol consumption. On admission, his height, weight, and body mass index were 180 cm, 77 kg, and 23, respectively. His vital signs at the time were blood pressure, 115/58 mmHg; heart rate, 80 bpm; respiratory rate, 22 breaths per min; SpO_2_, 92% on room air; and body temperature, 37.3 °C. Neurological examination revealed a Glasgow Coma Scale score of E4V4M5 and disorientation. The patient’s cranial nerves were intact, with no visual field defects, extraocular movement limitations, or neck stiffness. Right-sided arm weakness was noted (positive Barré’s sign), tendon reflexes were normal, and no sensory deficits or paresthesia were noted. Laboratory tests revealed a significant elevation in C-reactive protein levels, as well as increased blood urea nitrogen, creatinine, and D-dimer levels (Table 1).

Initial head CT demonstrated bilateral periventricular hypodensities, and the patient’s Evans index [10] was 0.34 (Figure 1). Mild enlargement of the Sylvian fissures was noted and the subarachnoid spaces over the high convexity appeared slightly narrowed, suggesting the absence of disproportionally enlarged subarachnoid space hydrocephalus (DESH) features. On hospital days 1 and 2, the four sets of blood cultures performed tested positive for Gram-negative rods, and treatment with intravenous meropenem was initiated for sepsis. Because of the patient’s elevated D-dimer levels and clinical presentation, transient ischemic attack or cerebral venous thrombosis were considered; however, echocardiography and lower extremity venous ultrasound revealed no evidence of thrombosis.

On day 2, brain magnetic resonance imaging (MRI) showed no signs of acute cerebral infarction; however, chronic lacunar infarcts were present, alongside periventricular white matter changes. No arterial or venous thrombosis, or stenosis, was observed (Figure 2).

On day 3, the causative organism was identified to be *Klebsiella pneumoniae*. Abdominal contrast-enhanced CT and urinalysis revealed liver abscess, urinary tract infection, and bacteremia. Based on antimicrobial susceptibility testing, treatment was de-escalated to ceftriaxone. Whole-spine MRI revealed cervical spinal canal stenosis between the C3–C6 levels. On day 9, a tap test was performed, and lumbar puncture revealed an opening pressure of 6 cm H_2_O. The CSF appeared clear with slight xanthochromism. A total drainage volume of 13 mL was achieved. Laboratory analysis revealed a glucose level of 115 mg/dL, a protein concentration of 38 mg/dL, and a white cell count of 2 cells/μL—findings which were inconsistent with meningitis. The absence of CSF pulsation and the low drainage volume raised the possibility of impaired CSF outflow because of cervical spinal canal stenosis and consequent communication disturbance. The patient’s gait disturbance improved significantly after the tap test was performed. Although his overall Mini-Mental State Examination (MMSE) score remained low, improvement was noted in the orientation component. Based on these findings, probable iNPH was diagnosed and a 12 mm brain abscess was incidentally found in the right basal ganglia (Figure 3).

By day 24, follow-up contrast-enhanced MRI confirmed encapsulation of the abscess (Figure 3). As the infection showed signs of improvement, a six-week course of ceftriaxone was planned before surgical intervention. On day 49, considering the possibility of impaired CSF circulation caused by cervical spinal canal stenosis and the presence of a right basal ganglia abscess, a left-sided ventriculoperitoneal (VP) shunt was placed. Postoperatively, the patient showed significant improvements in both his gait disturbance and cognitive function (Table 2). He soon regained ambulatory independence and was discharged.

### 2.2. Case 2

An 80-year-old man with a history of hypertension and diabetes mellitus, whose case was being managed at a nearby clinic, presented to our center with generalized muscle weakness. He had a several-year history of progressive gait disturbance, cognitive decline, and urinary incontinence. Two weeks prior to his admission, his frequency of falls increased significantly. Two days before admission, his urinary incontinence worsened, leading to a bedridden state.

His body temperature on arrival was 37.5 °C. Laboratory tests revealed an increased inflammatory response and mild dehydration (Table 3). Blood cultures were obtained, and bacteremia with *Staphylococcus haemolyticus* was confirmed. Intravenous antibiotic therapy was promptly initiated. Brain MRI revealed enlarged lateral ventricles and features consistent with DESH. The patient’s Evans index was 0.38, and his corpus callosum angle was measured at < 90°, supporting the diagnosis of iNPH acute cerebral infarction that was identified in the corona radiata, indicating the coexistence of hydrocephalus and ischemic stroke (Figure 4).

Initial management included the continuation of antibiotic therapy along with intravenous ozagrel sodium for antiplatelet effects, followed by the initiation of oral clopidogrel. A tap test was performed on day 21 of the patient’s hospitalization. His gait velocity improved from 1.255 to 1.339 steps/s thereafter, and his MMSE score increased from 11 to 16, suggesting responsiveness to CSF drainage. Lumber MRI showed spinal canal stenosis. On day 41, a right-sided VP shunt was inserted. Postoperatively, the patient demonstrated a significant improvement in independent ambulation (Table 4). Mild improvements in cognitive function and urinary symptoms were also observed. A follow-up CT performed on day 50 revealed a reduction in ventricular size (Figure 5).

By day 64, the patient’s overall symptoms and functional status had improved, and he was transferred to a rehabilitation hospital. At the time of writing this report, he is able to walk independently and has returned home with full independence in his daily activities.

## 3. Discussion

These two cases suggest a potential relationship between sepsis and the clinical manifestations or exacerbation of iNPH. In particular, systemic infection may induce changes in CSF dynamics, thereby contributing to the worsening of iNPH-related symptoms. One of the central mechanisms potentially involved is sepsis-associated encephalopathy (SAE), which is a diffuse brain dysfunction triggered by systemic inflammation in the absence of direct central nervous system infection [11,12]. SAE is increasingly recognized as a multifactorial condition involving neuroinflammation, microcirculatory failure, mitochondrial dysfunction, and blood–brain barrier (BBB) disruption [13]. Several biological processes associated with SAE may play roles in altering CSF physiology [14]. First, systemic inflammation can activate vascular endothelial cells and disrupt the BBB, allowing neurotoxic substances to enter the brain parenchyma. Second, cerebral autoregulation may be impaired, resulting in unstable cerebral perfusion. Third, mitochondrial dysfunction can lead to cellular energy failure and neuronal apoptosis. Finally, alterations in neurotransmitter synthesis and metabolism can occur as a result of increased protein catabolism and systemic inflammatory stress. In such cases, pre-existing chronic cerebral ischemia and latent CSF circulation disturbances may have been highlighted or exacerbated by the systemic effects of sepsis, resulting in an overt clinical presentation of iNPH. This interpretation is consistent with recent insights into the complex and dynamic nature of CSF absorption and circulation.

From a clinical standpoint, such cases highlight a critical point: when patients present with a combination of gait disturbance and ventricular enlargement in the setting of active infection, there is a risk that the diagnosis of iNPH may be overlooked. The possibility that infection-induced alterations in CSF dynamics contribute to the manifestation or worsening of hydrocephalus should be considered, particularly in older patients with comorbidities or chronic ischemic changes.

In Case 1, a 12 mm abscess was identified in the right basal ganglia. Although the abscess appeared to be encapsulated and not associated with acute neurological deficits at the time of tap test or surgery, it may have contributed to the local inflammatory milieu or altered CSF dynamics. The presence of the abscess complicates the causal interpretation of symptoms, and we acknowledge it as a potential confounder. In Case 2, although the MMSE score improved post-shunting, the FAB score decreased slightly. This discrepancy may reflect domain-specific cognitive fluctuations or measurement variability, and further follow-up would be necessary to determine clinical relevance.

Moreover, timely shunt surgery following appropriate antimicrobial and stroke treatments may lead to substantial clinical improvement in patients with hydrocephalus who develop acute ischemic stroke in the context of systemic infection. Although acute exacerbation of chronic neurological conditions triggered by systemic stressors is not uncommon, and clinical experiences suggest that events such as pneumonia, influenza, or surgery may precede the onset of iNPH symptoms, published reports describing this temporal association remain limited. Therefore, we consider this case series valuable in illustrating such a phenomenon with detailed radiographic and clinical follow-up. These findings reinforce the importance of recognizing hydrocephalus as a potential contributor to cerebral ischemia and also emphasize the need for personalized and multidisciplinary treatment strategies. Overall, these cases support the hypothesis that iNPH may emerge or worsen as a result of systemic inflammatory insults such as sepsis and that shunt surgery (when appropriately timed) can facilitate significant functional recovery.

## 4. Conclusions

These two cases represent rare clinical presentations in which the symptoms of iNPH became overt following systemic infections such as sepsis. We report these findings to highlight the possibility that similar mechanisms may be present in other patients with comparable courses and to provide insights for future clinical practice. In particular, patients with hydrocephalus who develop acute ischemic stroke caused by progressive cerebral ischemic damage may experience substantial symptomatic improvement following shunt surgery once the underlying infection and stroke have been appropriately treated. It is imperative to recognize the potential association between hydrocephalus and cerebral ischemia. Tailoring treatment strategies to account for this relationship may lead to better clinical outcomes in complex cases involving both systemic inflammatory conditions and disruptions to CSF circulation.

## Figures and Tables

**Figure 1 neurolint-17-00086-f001:**
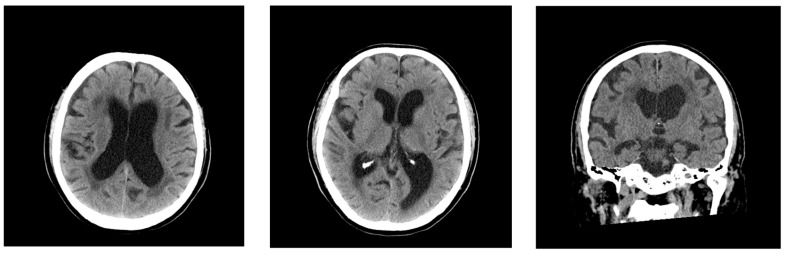
The head CT on admission in Case 1, showing bilateral periventricular hypodensities and ventricular enlargement. The patient’s Evans index was 0.34. His Sylvian fissures were mildly enlarged, and the subarachnoid spaces over the high convexity were slightly narrowed and did not exhibit typical DESH features.

**Figure 2 neurolint-17-00086-f002:**
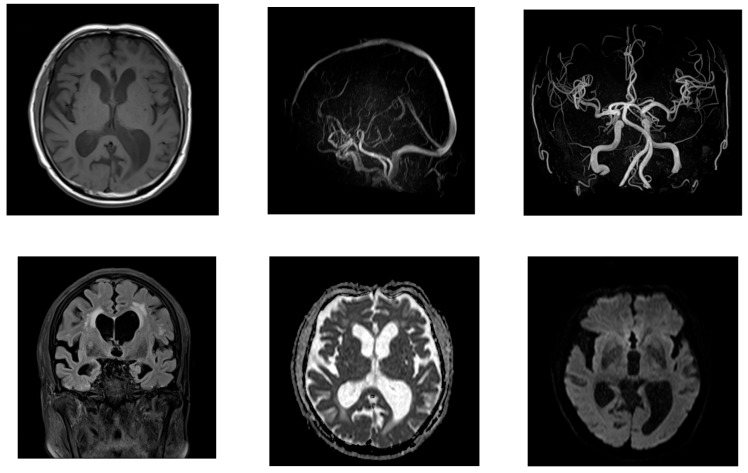
The MRI on day 2 of Case 1’s hospitalization, showing chronic ischemic changes. T2-weighted and FLAIR sequences revealed chronic lacunar infarcts and periventricular white matter hyperintensities. No acute ischemic lesions or vascular stenosis was observed.

**Figure 3 neurolint-17-00086-f003:**
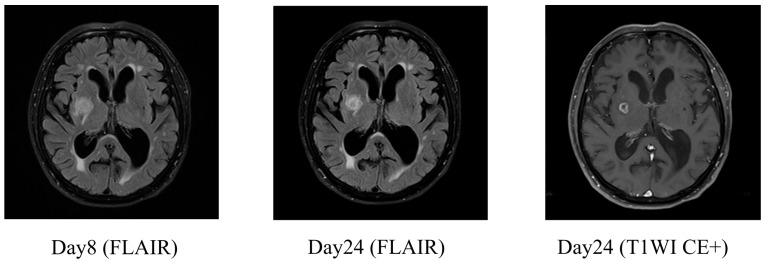
The brain MRI in Case 1 demonstrating a basal ganglia abscess and its encapsulation. The FLAIR image on day 8 showed a 12 mm lesion in the right basal ganglia, suggesting a brain abscess. The follow-up MRI on day 24 revealed clear encapsulation of the abscess on FLAIR and contrast-enhanced T1-weighted imaging (T1WI CE+).

**Figure 4 neurolint-17-00086-f004:**
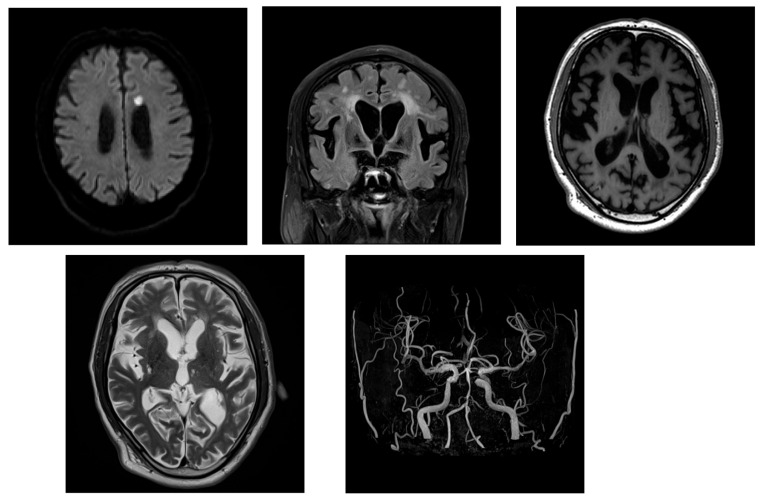
The MRI in Case 2 showing ventricular enlargement and acute cerebral infarction. The patient’s Evans index was 0.38, indicating DESH. Acute infarction was observed in the left corona radiata.

**Figure 5 neurolint-17-00086-f005:**
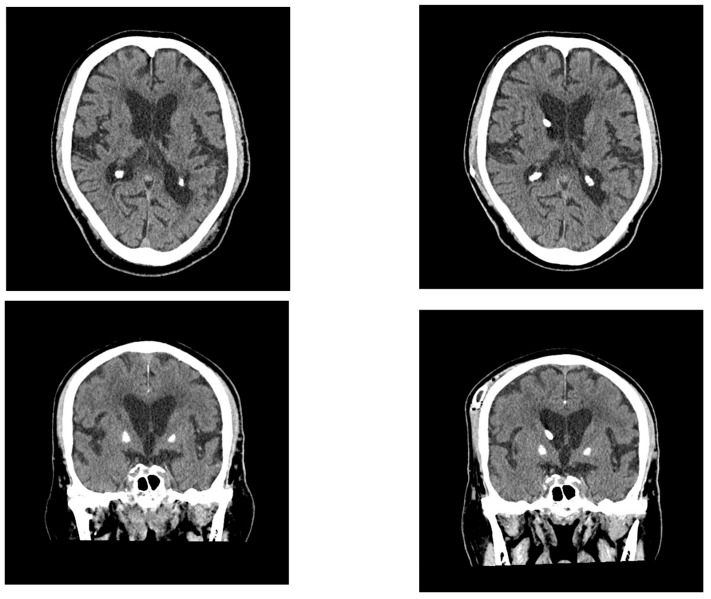
A follow-up head CT after shunt surgery in Case 2. The CT on postoperative day 9 showed a reduction in ventricular size vs. baseline, indicating a favorable radiological response to VP shunting.

**Table 1 neurolint-17-00086-t001:** Laboratory test on admission in Case 1.

Hematological Tests		
WBC	10.6	×10^9^/L
Hb	11.7	g/dL
Plt	96	10^9^/L
**Biochemical testing**		
ALB	2.7	g/dL
BUN	51	mg/dL
Cre	1.66	mg/dL
ALP	227	IU/L
AST	59	IU/L
ALT	68	IU/L
LDH	1282	IU/L
T-Bil	0.9	mg/dL
CRP	19.19	mg/dL
Glu	234	mg/dL
Na	134	mmol/L
K	3.8	mmol/L
Cl	98	mmol/L
**Coagulation System Tests**		
D-dimar	7.1	μg/dL

**Table 2 neurolint-17-00086-t002:** Results of cognitive and gait assessments in Case 1.

		Pre-Tap Test	Post-Tap Test **	Post-Shunting
TUG	time (s)	44.2	29.3	21.27
	step (step)	103	71	－
10 m walking *	time (s)	－	－	9.52
	step (step)			19.5
MMSE		22/30	21/30 ***	27/30
FAB		10/18	11/18	16/18

TUG, Timed Up and Go test; MMSE, Mini-Mental State Examination; FAB, Frontal Assessment Battery; * TUG was difficult to perform at the level of walking with a walker before shunting; therefore, it was performed using 10 m walking with a walker as a reference; ** one week after the tap test; *** the orientation items showed signs of improvement.

**Table 3 neurolint-17-00086-t003:** Laboratory test on admission in Case 2.

Hematological Tests		
WBC	12.4	×10^9^/L
Hb	17.0	g/dL
Plt	244	10^9^/L
**Biochemical testing**		
ALB	3.8	g/dL
BUN	22	mg/dL
Cre	1.18	mg/dL
ALP	213	IU/L
AST	34	IU/L
ALT	33	IU/L
LDH	225	IU/L
T-Bil	1.3	mg/dL
CRP	9.11	mg/dL
Glu	174	mg/dL
Na	145	mmol/L
K	3.6	mmol/L
Cl	108	mmol/L
**Coagulation System Tests**		
D-dimar	—	μg/dL

**Table 4 neurolint-17-00086-t004:** Results of cognitive and gait assessments in Case 2.

		Pre-Tap Test	Post-Tap Test **	Post-Shunting
TUG	time (s)	－	－	18.74
	step (step)			
10 m walking *	time (s)	21.52	17.92	11.95
	step (step)	27	24	21
MMSE		11/30	16/30	15/30
FAB		4/18	8/18	6/18

TUG, Timed Up and Go test; MMSE, Mini-Mental State Examination; FAB, Frontal Assessment Battery; * TUG was difficult to perform at the level of walking with a walker; therefore, it was performed using 10 m walking with a walker as a reference; ** one day after the tap test.

## Data Availability

The data supporting the findings of this study are available from the corresponding author upon reasonable request. However, due to privacy and ethical restrictions, raw patient data are not publicly available.

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
