# Peer review of "Idiopathic Normal-Pressure Hydrocephalus Revealed by Systemic Infection: Clinical Observations of Two Cases"

_2035-8377, 2025, doi:10.3390/neurolint17060086_

Round 1
Reviewer 1 Report
Comments and Suggestions for Authors
1. The data expressions of MMSE and FAB in Table1 and Table2 are not consistent.(10/18, 11/18, 16)
2. In Case 1, the presence of Basal ganglia abscess may also have an impact on neurological symptoms and CSF dynamics, and it is suggested that a paragraph on its possible contribution should be added to the discussion section to avoid confusion about cause and effect.
3. Although the MMSE of Case 2 was slightly improved, the FAB score dropped from 8 to 6 points, which may indicate that some cognitive function has not yet recovered. It is worth exploring whether the data is accidental.
4. In the introduction part, the manuscript effectively explored the existing literature on the mechanism of hydrocephalus and laid a solid theoretical foundation for the discussion of this case. It is believed that glymphatic system function impairment plays an important role in the pathogenesis of iNPH, and it would be more interesting if the authors could explore more literature on hydrocephalus and glymphatic system: PMID: 31959516, PMID: 33488070.
Author Response
- The data expressions of MMSE and FAB in Table1 and Table2 are not consistent.(10/18, 11/18, 16)
Our response:
Thank you very much for your comment regarding the inconsistency in the data expressions of MMSE and FAB in Table 1 and Table 2.
We have carefully reviewed and corrected the formatting of MMSE and FAB scores to ensure consistency across all tables. Specifically, the expressions in Table 2 and Table 4 have been revised and now follow a unified format (e.g., score/full score such as 11/30 or 8/18).
We appreciate your attention to detail.
- In Case 1, the presence of Basal ganglia abscess may also have an impact on neurological symptoms and CSF dynamics, and it is suggested that a paragraph on its possible contribution should be added to the discussion section to avoid confusion about cause and effect.
Our response:
Thank you for your thoughtful comment regarding the basal ganglia abscess in Case 1.
We agree that the presence of the abscess could potentially have influenced neurological symptoms and CSF dynamics. In response to your suggestion, we have added a paragraph in the Discussion section acknowledging this possibility and noting it as a potential confounding factor.
We appreciate your helpful feedback in clarifying this aspect of the case.
- Although the MMSE of Case 2 was slightly improved, the FAB score dropped from 8 to 6 points, which may indicate that some cognitive function has not yet recovered. It is worth exploring whether the data is accidental.
Our response:
Thank you very much for your insightful observation.
We have added a brief discussion of the discrepancy between MMSE improvement and the slight decrease in FAB score in Case 2. We acknowledge that this finding may reflect variability in cognitive domains or measurement error, and further follow-up would be needed to determine its clinical significance.
We appreciate your suggestion, which allowed us to better contextualize the cognitive findings.
- In the introduction part, the manuscript effectively explored the existing literature on the mechanism of hydrocephalus and laid a solid theoretical foundation for the discussion of this case. It is believed that glymphatic system function impairment plays an important role in the pathogenesis of iNPH, and it would be more interesting if the authors could explore more literature on hydrocephalus and glymphatic system: PMID: 31959516, PMID: 33488070.
Our response:
Thank you very much for your helpful suggestion regarding additional literature on the glymphatic system and its relevance to the pathogenesis of idiopathic normal pressure hydrocephalus (iNPH).
After reviewing the references you kindly suggested (PMID: 31959516 and PMID: 33488070), we found that they provide valuable insights into glymphatic dysfunction and its potential contribution to iNPH. We have therefore incorporated both references into the revised manuscript (Introduction, lines 80–83), as they align well with our discussion of CSF circulation impairment in the context of systemic inflammation and hydrocephalus.
We would like to emphasize that the decision to include these references was made not solely on the basis of your suggestion, but after careful evaluation of the literature and relevance to the conceptual framework of our case series.
We appreciate your guidance, which helped enhance the scientific foundation of our introduction.
Reviewer 2 Report
Comments and Suggestions for Authors
The authors present two cases of elderly individuals with sepsis and the development of NPH symptoms during the infection. Their case histories are well laid out and both diagnostic testing and response to shunting is indeed compatible with normal pressure hydrocephalus.
My only comment and criticism is the assumption that this is a rare event. Although it may not be well published in the literature, acute exacerbation of chronic neurologic disease with a new"stressor"is not an unusual phenomena. With NPH in particular, patients will often note a significant medical event coincident with their onset of symptoms. Pneumonia, influenza, coronary artery bypass, TIA, etc. have all been" initiating "events in my experience for patients describing the triad of symptoms with NPH. Furthermore, the coincident radiographic evidence of old cerebrovascular disease in this population is relatively common.
Author Response
The authors present two cases of elderly individuals with sepsis and the development of NPH symptoms during the infection. Their case histories are well laid out and both diagnostic testing and response to shunting is indeed compatible with normal pressure hydrocephalus.
My only comment and criticism is the assumption that this is a rare event. Although it may not be well published in the literature, acute exacerbation of chronic neurologic disease with a new"stressor"is not an unusual phenomena. With NPH in particular, patients will often note a significant medical event coincident with their onset of symptoms. Pneumonia, influenza, coronary artery bypass, TIA, etc. have all been" initiating "events in my experience for patients describing the triad of symptoms with NPH. Furthermore, the coincident radiographic evidence of old cerebrovascular disease in this population is relatively common.
Our response:
Thank you very much for your valuable comment.
We agree with your observation that acute exacerbation of chronic neurological conditions by systemic stressors is not uncommon in clinical practice, and that events such as pneumonia, influenza, or surgical procedures may trigger the manifestation of iNPH symptoms. In light of your suggestion, we have revised the Discussion section to reflect this perspective and acknowledge that such associations, while possibly underreported in the literature, are frequently encountered in clinical experience.
Specifically, we have added the following sentence to the final part of the Discussion:
“Although acute exacerbation of chronic neurological conditions triggered by systemic stressors is not uncommon, and clinical experiences suggest that events such as pneumonia, influenza, or surgery may precede the onset of iNPH symptoms, published reports describing this temporal association remain limited. Therefore, we consider this case series valuable in illustrating such a phenomenon with detailed radiographic and clinical follow-up.”
We appreciate your insightful input, which helped improve the balance and clarity of our discussion.